# Biomedical Applications of Hyaluronic Acid-Based Nanomaterials in Hyperthermic Cancer Therapy

**DOI:** 10.3390/pharmaceutics11070306

**Published:** 2019-07-01

**Authors:** Subin Kim, Myeong ju Moon, Suchithra Poilil Surendran, Yong Yeon Jeong

**Affiliations:** 1Department of Biomedical Sciences, Biomolecular Theranostics (BiT) Lab, Chonnam National University Medical School, Hwasun 58128, Korea; 2Department of Radiology, Biomolecular Theranostics (BiT) Lab, Chonnam National University Medical School, Hwasun 58128, Korea

**Keywords:** hyaluronic acid (HA), nanomaterials, cancer, hyperthermia, photothermal therapy (PTT), magnetic hyperthermia therapy (MHT), combined cancer treatment

## Abstract

Hyaluronic acid (HA) is a non-sulfated polysaccharide polymer with the properties of biodegradability, biocompatibility, and non-toxicity. Additionally, HA specifically binds to certain receptors that are over-expressed in cancer cells. To maximize the effect of drug delivery and cancer treatment, diverse types of nanomaterials have been developed. HA-based nanomaterials, including micelles, polymersomes, hydrogels, and nanoparticles, play a critical role in efficient drug delivery and cancer treatment. Hyperthermic cancer treatment using HA-based nanomaterials has attracted attention as an efficient cancer treatment approach. In this paper, the biomedical applications of HA-based nanomaterials in hyperthermic cancer treatment and combined therapies are summarized. HA-based nanomaterials may become a representative platform in hyperthermic cancer treatment.

## 1. Introduction

Over the years, a large amount of effort has been devoted towards treating cancers. Advances in nanotechnology, particularly in the field of drug nanocarriers, have enabled biocompatible and effective drug delivery to cancer sites with reduced side-effects [1]. The development of nanoparticles and nanotechnology has elicited significant and precise cancer diagnoses and treatment tools [2]. Locally concentrated drugs in tumor regions are critically required, while other organs or tissues need to receive low doses of the drugs in order to avoid severe side-effects [3]. To distribute a drug into a tumor site while avoiding other organs, it is necessary to design specifically engineered nanoparticles by modifying the surface with targeting moieties [4,5]. Many researchers have designed nanoparticles with specific ligands (called active targeting ligands) which have further improved the efficacy of drug delivery to specific tumor sites [6,7].

One of the main active targeting moieties is hyaluronic acid (HA). HA is a potentially useful polymer, due to its biodegradable, biocompatible, non-inflammatory, and non-toxic properties [8]. As with other naturally occurring polysaccharides, HA has been reported to be highly biodegradable. In addition, as it is a polysaccharide, its biocompatibility also makes it unique, when compared to other polymeric systems. HA has been shown to not elicit any immune response in the body and can be used as an efficient delivery system for cancer treatment [9]. The intracellular signaling of HA occurs through its binding to certain receptors, such as CD44, RHAMM, and LYVE-1 [10]. CD44, a major cell adhesion receptor of HA, is abundantly found (relative to normal cells) in the tumor extracellular matrix [11]. Thus, HA can selectively target cancerous cells over-expressing CD44 and many studies have addressed HA-modified nanomaterials as cancer targeting moieties to enhance cancer therapy without side-effects [12,13,14,15,16]. This CD44 targeting is a unique property of HA-based nanomaterials. Thus, HA could be an ideal biomaterial for CD44-over-expressing cancer cells.

HA is widely distributed throughout the body, in such places as the extracellular matrix and synovial fluids and is mainly degraded by enzymatic process. Three different types of enzymes participate in HA degradation: Hyaluronidase, b-d-glucuronidase, and β-*N*-acetyl-hexosaminidase. Firstly, HA is degraded by hyaluronidase by splitting (high molecular weight) HA into small oligosaccharides. Subsequently, the involvement of b-d-glucuronidase and β-*N*-acetyl-hexosaminidase helps to degrade a remaining oligosaccharide [17]. The high biocompatibility, CD44 targeting ability, and enzyme degradability make HA nanomaterials unique from other nanomaterials. Due to the biodegradability, biocompatibility, and non-toxicity of HA, diverse nanomaterial structures involving HA, such as micelles, polymersomes, hydrogels, and nanoparticles, could be designed for the treatment of cancer.

Hyperthermia, one tool for cancer treatment, aims to increase the temperature at the tumor site and, consequently, cause cancer cell death [18]. For the therapeutic efficacy of hyperthermia in cancer treatment, the targeted tumor regions must reach temperatures in the range of 42–46 °C [19]. When cells begin to be exposed to this range of temperatures, protein denaturation occurs, inducing a large proportion of co-aggregated denatured proteins [18]. Moreover, the increasing temperature affects the function of the cellular structure and changes the intracellular functions, which finally leads to cancer death [20]. Due to the sudden variation of temperature in cancerous cells, insufficient blood flow, nutrition, and oxygen supply are caused within the blood vessels in the tumor environment; however, tumors have more tolerance to temperature changes [21,22]. Hyperthermia has been applied in cancer treatment to maximize the therapeutic efficacy against cancer. Photothermal therapy (PTT) and magnetic hyperthermia therapy (MHT) are representative hyperthermia-mediated cancer treatments Figure 1. Due to the great capability of HA to target CD44-over-expressing cancer cells, HA-based nanomaterials used in PTT and MHT could accumulate in CD44-over-expressing cancer cells and, subsequently, elicit an anti-cancer effect.

In this review, we mainly focus on HA-modified nanomaterials for hyperthermic cancer treatment. In addition, combined cancer therapy with hyperthermia will be discussed in detail.

## 2. Photothermal Therapy

PTT is one of the hyperthermia-mediated cancer treatments. PTT is novel, safe, effective, and less invasive, which operates by increasing the temperature of cancerous cells and eliminating tumors with minimal damage to adjacent normal cells [23,24,25]. The absorbed photothermal energy through photothermal agents are converted into heat, which causes cancer cells become susceptible to the induction of apoptosis [26,27]. Photothermal agents, such as near-infrared (NIR) fluorescence or photoabsorbers, are widely used in PTT to generate heat within the local tumor micro-environment [28]. Plasmonic nanoparticles, including gold nanoparticles (AuNPs), graphene oxide (GO), and prussian blue nanoparticles (PB NPs), are also well-known as photothermal converting nanomaterials for hyperthermia [29]. A recent study reported a manganese phthalocyanine (MnPc) nanosheet as a photoactive ligand, which exhibited a photothermal conversion efficiency of 72.3% [30].

### 2.1. NIR-Loaded Nanoparticles

To generate heat at the tumor site, NIR fluorescence dye is extensively used as a contrast agent in PTT Table 1. NIR dye-based nanomaterials have shown excellent thermal ablation upon NIR laser irradiation [31]. Among a diverse range of NIR dyes, indocyanine green (ICG) is one of the representative NIR dyes which have been approved by the Food and Drug Administration (FDA). Moreover, ICG generates cytotoxic ROS under NIR laser irradiation [32]. However, due to the poor water solubility of NIR dyes, several studies have endeavored to successfully deliver NIR dyes to cancer cells for PTT. HA-based nanomaterials could be potential nanocarriers for delivering NIR dyes to the tumor site. Micelles, polymersomes, and hydrogels have been utilized as HA-based nanocarriers and NIR-responsive materials, such as NIR dyes, gold nanoparticles, graphene oxide nanoparticles, prussian blue nanoparticles, and magnetic nanoparticles, have been widely used for cancer treatment, with the hope of having great targeting ability and water solubility Figure 2 [33].

IR–780-loaded HA-based micelles (HA–IR–780) were synthesized by a simple dialysis method for PTT effects in TC-1 lung cancer cells [12]. Upon laser irradiation of the tumor site, the temperature in the tumors treated in the HA–IR–780 group reached 49.9 °C, which led to significant tumor reduction when compared to the control group, without any side-effects. Another group developed self-assembled tumor-targeting HA–IR–780 nanoparticles for PTT in a CD44-over-expressed orthotopic bladder cancer model. This group showed a maximum temperature of 48.1 ± 1.81 °C in the tumors, contributing to improvements in bladder cancer treatment [34]. NIR-conjugated HA nanoparticles (HA, Mw = 50 kDa) have been synthesized for PTT, demonstrating superior anti-cancer effects and an improved life span, compared to the control group [35]. NIR dye-conjugated HA NPs and an encapsulation of perfluorooctylbromide (PFOB) NPs (PFOB@IR825–HA–Cy5.5) were synthesized for enhanced PTT. These NPs could selectively target CD44-over-expressing HT-29 human colon cancer cells. The tumor temperature, after injection with the NPs, reached approximately 50 °C, showing a great ability of these NPs to destroy tumor cells. Furthermore, significant tumor volume reduction was observed with laser application, without any side-effects, in major normal organs [36]. Thus, NIR dye-conjugated HA-based nanomaterials can be used as efficient theranostic agents.

### 2.2. Gold Nanoparticles (AuNPs or GNRs)

AuNPs have been extensively studied for both molecular imaging and PTT [37]. Compared with other nanostructures, AuNPs have several advantages, including easy synthesis methods and surface modification, along with biocompatibility, and these characteristics make AuNPs easy to apply in the diagnosis and treatment of cancer [38,39]. More importantly, AuNPs have shown efficient local heating upon excitation of surface plasmon oscillation [40]. When the wavelengths of the AuNPs match with the wavelength of the NIR laser, the AuNPs absorb the NIR laser and convert it into heat at a high efficiency rate. Thus, AuNPs are mostly utilized as photothermal agents for cancer treatment. Diverse gold nanostructures, including gold nanorods (GNRs), gold nanoshells, gold nanocages, and carbon nanotubes, have also been demonstrated to have potential for use in PTT [38,41,42]. HA-modified AuNPs enhance multi-functional theranostics for imaging and PTT Table 1.

Wang et al. [23] reported a rationally designed nanosystem based on a gold nanostar/siRNA against HSP72 (siHSP72) of HA for selectively sensitizing MDA–MB–231 cancer cells. These nanosystems were synthesized by a surfactant-free method and HA with a molecular weight of 10 KDa was used. In comparison to the case without an HA coating, GNS/siHSP72/HA showed improved pharmacokinetics, with high accumulation in the tumor. These NPs could selectively target cancer cells by CD44 endocytosis and could have an anti-cancer effect after laser irradiation. Other groups have synthesized HA-modified Fe_3_O_4_@Au core/shell nanostars for tri-mode imaging and PTT in HeLa cancer cells [42]. An excellent anti-cancer effect was proven in a HeLa tumor-bearing mice model, showing complete tumor removal. After laser irradiation, the NPs showed a long lifespan, significantly higher than the control group. HA played an important role in eliciting an effective photothermal effect. HA-conjugated AuNP-coated poly(glycidylmethacrylate) nanocomposites (PGMA@Au–HA) were fabricated for PTT. The selective targeting ability and great photothermal efficacy of these NPs were demonstrated in cell viability study. After laser irradiation, high cell cytotoxicity was observed in CD44-over-expressing SCC cancer cells, while low cell cytotoxicity was shown in HaCaT normal cells [43].

High or low molecular weight HA-coated gold nanobipyramids (GBPs@h–HA and GBPs@l–HA, respectively) were synthesized for PTT in MDA–MB–231/Luc cells. GBPs@h–HA showed superior targeting ability, when compared with GBPs@l–HA by HA–CD44 endocytosis. Upon 808 nm laser irradiation, a higher therapeutic efficacy was observed in the group GBPs@h–HA, compared to GBPs@l–HA, in both in vitro and in vivo studies [44]. The molecular weight of HA plays a key role in targeting and photothermal efficacy in CD44-over-expressing cancerous cells.

Recently, platinum (Pt)-modified GNRs have been developed as attractive materials for scavenging ROS [45,46]. ROS generation by PTT could cause the irreversible damage to DNA, proteins, and mitochondria, which may lead to cellular dysfunction in healthy tissues [47,48]. One group showed the synergistic efficacy of Pt-coated GNRs by demonstrating the photothermal efficacy and ROS-scavenging activity of these NPs [46]. Pt-modified GNRs could be a potential candidate to improve the photothermal efficacy in cancer, while avoiding severe side-effects in undesired tissues.

### 2.3. Functionalized Graphene Oxide (GO)

GO has been shown to be effective in biomedical applications, including biosensing, protein delivery, gene delivery, and drug delivery. GO exhibits non-toxicity at certain concentrations, degrades easily, and can have improved water solubility by modification of the hydrophilic functional groups in GO. Additionally, GO has a high surface area, good NIR absorbance, facile synthesis, and abundant chemical functional groups, which are appropriate for the loading of hydrophilic and hydrophobic materials [27,49,50]. However, the property in NIR light absorption of GO is much lower than reduced graphene oxide (RGO) [51,52,53]. RGO is well-known as an effective NIR-responsive nanomaterial which generates heat in a localized environment, leading to irreversible cell and tissue damage [54]. HA-functionalized GO could be a potential efficient target and an optimal material as a PTT agent Table 1. One group [55] developed HA functionalized-reduced graphene oxide (rGO) for targeted cancer PTT. The photothermal efficacy of HA-rGO has been proven by the increased temperature of HA-rGO to 33 °C at its highest concentration (75 μg/mL of rGO), a temperature that is sufficient to kill MCF-7 human breast cancer cells.

Nanographene oxide (NGO) is another GO-mediated nanomaterial which is a promising photothermal agent for PTT, due to its high photothermal responsiveness and low toxicity [56,57]. Principally, Xiong et al. [58] have explored the use of NGO in PTT. They prepared a PEG-modified NGO by cleavable disulfide linkages (NGO–SS–PEG) The temperature of NGO–SS–PEG reached approximately 70 °C with laser irradiation, demonstrating great photothermal efficacy. A NGO–HA conjugate (NGO–HA) was developed for PTT of melanoma skin cancer [59]. The anti-tumor efficacy of NGO–HA was proven in a tumor volume study, demonstrating the superior anti-cancer effect of NGO–HA with NIR laser irradiation in a melanoma tumor-bearing mice model.

### 2.4. Prussian Blue Nanoparticles (PB NPs)

PB NPs have been recently described as a new material for photothermal agents in cancer treatment, due to a superior NIR photothermal conversion effect [60,61]. Most importantly, PB NPs have been approved by the US FDA for the clinical treatment of radioactive exposure. Furthermore, due to their biocompatibility, biodegradability, and ease of synthesis, PB NPs have been utilized in biomedical applications [62]. For example, Fe_3_O_4_@PB NPs have been developed by growing PB nanoshells around Fe_3_O_4_ nanocores (Fe_3_O_4_@PB NPs) for targeted PTT [63]. After 808 nm laser irradiation, the temperature of a tumor injected with Fe_3_O_4_@PB NPs was elevated to over 60 °C. Moreover, a superior anti-cancer effect was observed in treatment using Fe_3_O_4_@PB NPs. Therefore, PB NPs are one of the emerging new nanomaterials for PTT.

HA-conjugated PB NPs have been used for targeted cancer imaging and therapy Table 1. HA-conjugated magnetic PB@quantum dots NPs have been developed for targeted PTT. These NPs were mainly accumulated at tumor site by HA-CD44 endocytosis and an external magnetic field (MF). The temperature in the tumor region of mice injected with NPs and treated with a MF reached 49 °C. Furthermore, significant tumor volume reduction was observed in the NPs with MF group, without severe side-effects [64]. Such nanomaterials provide potential candidates for targeted PTT.

## 3. Magnetic Hyperthermia Treatment

MHT, also known as magnetic fluid hyperthermia (MFH) or alternating magnetic field (AMF) treatment, is a type of cancer therapy which combines specific materials to absorb external magnetic energy and convert it into heat [65]. To maximize the effect of MHT-based cancer therapy, magnetic nanoparticles (MNPs) are mostly utilized. Iron oxide nanoparticles are a representative magnetic nanomaterial for MHT cancer treatment, due to their biocompatibility, high magnetic susceptibility, and superparamagnetic behavior [66]. The most commonly utilized MNPs are magnetite (Fe_3_O_4_), maghemite (γ-Fe_2_O_3_), and hematite (α-Fe_2_O_3_) [67]. In MHT, heat is produced after the localization of MNPs under the application of an AMF [68]. There are two main mechanisms of MNPs for heat generation: Hysteresis loss and relaxational losses. In hysteresis loss, multi-domain MNPs follow the same direction of the external magnetic moment. In relaxational losses, single-domain superparamagnetic NPs generate heat by Néel relaxation and Brownian relaxation [18].

Due to the biocompatibility and relatively low cytotoxicity of Fe_3_O_4_, it has been applied in many biomedical applications, including magnetic resonance imaging (MRI), computed tomography (CT), and hyperthermic cancer treatment [69,70]. For example, Ohtake et al. [71] synthesized MNP-based nanomaterials for MHT. The hyperthermic efficacy of these NPs was evaluated in a mouse leg tumor model, where the temperature of a tumor injected with these NPs reached 43.1 °C. The combination of these NPs and AMF showed great anti-cancer effects, as compared to single NPs treatment. One group [72] reported HA-modified magnetic iron oxide nanoparticles (HA-Fe_3_O_4_) for the MR imaging of endometriosis lesions in rats, showing improved visualization of endometriosis at 2 h post injection.

Several decades ago, various engineered superparamagnetic iron oxide NPs (SPIONs) were tested as promising candidates for cancer treatment. Recently, cobalt NPs have attracted attention as interesting chemical components in biomedical applications, due to their size and shape, catalysis, and magnetism [73]. It is important to note that cobalt possesses magnetic properties approximately twice as strong as MNPs [74]. Surface-modified cobalt oxide NPs show great in vitro targeting ability and anti-cancer effects [73]. It has been suggested that optimizing MNPs, in order to increase their thermal efficacy, particle size, and coating materials, should be considered [75]. In particular, coating is one solution for improving the function of MNPs, resulting in good dispersion, high stability, and sufficient loading of drugs [76]. By applying coatings on the surface of MNPs, large-sized clusters can be avoided, which are not appropriate for in vivo applications; thus, colloidal stability and a sufficient dose of MNPs can be attained at the tumor site [77]. HA-modified MNPs provide stable dispersibility and cytocompatibility for biomedical applications [78].

Several studies have addressed various polymer-coated MNPs for MHT [79,80,81,82,83]. However, few studies have explored HA-modified nanomaterials for MHT in the treatment of cancer. Our group [84] fabricated HA (Mw = 6800 Da) and polyethylene glycol-conjugated superparamagnetic iron oxide nanoparticles (HA–PEG10–SPIONs) for AMF-mediated hyperthermia. The heating ability of HA–PEG10–SPIONs with AMF showed a hyperthermic temperature of approximately 42 °C. In vitro hyperthermia cell viability studies exhibited marked cell death by HA–CD44 receptor mediated targeting. In contrast, a hyperthermic cell viability study in NIH3T3 cells did not show cell death, due to the selective uptake of HA. HA-conjugated SPIONs have potential as MHT-based hyperthermic agents in SCC7 cancer cells.

HA-tethered FePt alloy NPs have been designed for multi-modal therapy in glioblastoma cancer cells [85]. In this study, due to the superior magnetic properties of FePt, magnetic hyperthermia led to a 3–4 °C increase in cellular temperature, demonstrating a better heating ability. HA-coated lanthanum strontium manganates (LSMO) have been studied with regards to their morphology, magnetic properties, and heating efficacy under AMF with different concentrations of each element. The HA-coated La_0.7_Sr_0.3_MnO_3_ MNPs exhibited a heating temperature of 45.7 °C and a magnetic particle size of approximately 100 nm, indicating that they are suitable MNPs for mild hyperthermia [86].

In the clinic, a company using the AMF system-based MagForce AG was established in Germany. Phase 1 and 2 clinical studies for patients with prostate cancer were performed under the MagForce AG using MNPs by intratumoral injection, showing executable approach outcomes. Furthermore, the phase 2 clinical study for patients with intermediate prostate cancer demonstrated an improved median survival of 13.4 months, which prevented acute toxicity from the beginning of tumor recurrence [18]. Clinical applications of AMF system-based MHT is a feasible cancer treatment in the future.

## 4. Combined Therapies Using Hyperthermia

To combat cancer with early treatment and maximize therapeutic efficacy against cancer, hyperthermia has been utilized as an adjuvant to overcome the limitations of single cancer treatment modalities [87]. Hyperthermia improves chemotherapy through enhanced tissue perfusion, further triggering the absorption of chemotherapeutic drugs into cells. Furthermore, combined hyperthermia and radiotherapy has become an attractive cancer treatment. The formation of cytotoxic free radicals from radiotherapy induces protein denaturation and DNA damages in cancer cells, while hyperthermia blocks the DNA repair system [88]. Detailed actions of combined therapies involving hyperthermia are discussed in below.

### 4.1. Photothermal Chemotherapy

Combining hyperthermia and chemotherapy could be an efficient treatment approach to overcome side-effects and improve the anti-cancer effects. Hyperthermia by NIR laser irradiation facilitates the shrinkage of blood vessels and tumor ablation, after which, the release of chemotherapeutic drugs by NIR-induced hyperthermia could inhibit the growth of cancers [89]. The mechanism of combined hyperthermia and chemotherapy is associated with membrane damage, cytoskeleton injury, and DNA repair [90]. Elevated uptake of chemotherapeutic drugs into cancer cells can be attributed to increased cancer cell membrane permeability and, finally, DNA damage is increased [91]. Photothermal chemotherapy has become a practical cancer treatment in a synergistic manner Figure 3. In combined cancer treatment, HA could act as a targeting moiety for cancer cells which possess an associated CD44 receptor expressed on the cell surface, where more than one therapeutic agent could be loaded or conjugated to the HA [33]. Therefore, HA-based nanomaterials enable a combined cancer treatment. HA-modified nanoparticles have potential in synergistic photothermal chemotherapy.

NIR dyes are promising photothermal agents for hyperthermia. Due to the deep tissue-penetrating properties of NIR light, NIR dyes have been widely utilized for cancer imaging and treatment [92]. One group [93] developed HA-based NPs with doxorubicin (DOX) and ICG for photothermal chemotherapy. The improved cytotoxic activity of these NPs was proven in CD44-over-expressing HCT-116 cancer cells. The temperature in tumors treated with these NPs reached 47.7 °C, which was enough to induce irreversible tissue damages. Furthermore, these NPs demonstrated improved in vivo anti-cancer effects, compared to either PTT or chemotherapy alone. Another researcher reported IR780- and DOX-loaded HA-based NPs (IR/DOX–HPN) for combined cancer therapy. Due to selective targeting recognition of HA, these NPs were successfully accumulated to CD44-over-expressing MCF-7 cancer cells, compared to NHDF normal cells. In addition, the cell viability study of these NPs with laser irradiation showed considerable cytotoxicity against MCF-7 cells [94]. Thus, PTT agent- and chemotherapeutic drug-loaded nanomaterials are promising for combined therapy.

GNRs have been commonly used in PTT, due to their strong surface plasmon resonance absorption in the NIR region [95]. DOX-conjugated HA-functionalized folate GNRs have been fabricated for photothermal chemotherapy [96]. The tumors of MCF-7 tumor-bearing mice treated with combined photothermal chemotherapy showed complete tumor inhibition within 20 days post-injection, while treatment with a single modality showed only moderate anti-cancer effects. Recently, GNRs encapsulated in mesoporous silica (mSiO_2_) have been shown high drug loading efficiency, biocompatibility, and heat conversion capability [97]. GNRs@mSiO2–HA–RGD (RGD: Arginine–glycine–aspartate) nanocarriers have been synthesized for targeting CD44- and integrin receptor-over-expressing cancer cells (SKOV-3). The NPs exhibited synergistic therapeutic efficacy, compared to that of chemotherapy or PTT alone [98]. Another group [99] also synthesized DOX-loaded HA and RGD-conjugated mesoporous silica-coated GNRs. The cell viability study demonstrated that combined therapy showed a better therapeutic efficacy, compared to that of single treatment modalities. GNR-loaded HA nanogels showed synergistic therapeutic efficacy against MCF-7 ADR cancer cells [100]. With the help of GNR-induced hyperthermia, the GNR-loaded HA nanogels could increase the concentration of DOX in MCF-7 ADR cells, consequently inducing cancer apoptosis Table 2.

One group reported HA-modified and chemotherapeutic drug-loaded GO for targeted drug delivery and therapy. This group successfully loaded DOX and PTX onto GO–HA, showing a significant cancer killing effect by MHT in CD44-over-expressing MDA–MB–231 cancer cells, but not in BT–474 cells [101]. In relation to functionalized GO, a GO- and DOX-conjugated HA nanogel (GDH nanogel) was fabricated for combined PTT and chemotherapy. The accumulative release of DOX from the GDH nanogel was significantly increased at pH 5.0 with laser irradiation, compared to that at pH 7.4 conditions. The GDH nanogel was preferentially accumulated in CD44-over-expressing A549 cancer cells. The tumor temperature after injection with the GDH nanogel was elevated to about 52 °C, which further induced superior anti-cancer effects [102]. Due to the photothermal efficacy and high drug-loading capacity of functionalized GO, the GDH nanogel could be an ideal candidate for combined PTT chemotherapy.

Docetaxel-loaded polypyrrole (PPY) and HA-modified phospholipid nanoparticles (DTX/PPN@PPY@HA) have been used for photothermal chemotherapy. Polypyrrole nanoparticles (PPY) are one of the potential therapeutic polymers for photo-acoustic imaging and PTT, due to their good biocompatibility, low toxicity, strong NIR absorbance, and efficiency of photothermal conversion [103,104]. With laser irradiation of a 4T1 tumor-bearing mice model, complete tumor inhibition was observed in the group treated with DTX/PPN@PPY@HA after laser irradiation. Due to good cancer-targeting effects and photothermal efficacy, DTX/PPN@PPY@HA showed good synergistic anti-cancer effects in photothermal chemotherapy [105].

DOX- and HA-decorated graphene oxide nanosheets (HSG–DOX) have been synthesized and successfully accumulated into CD44-over-expressing MDA–MB–231 cancer cells by the enhanced permeability and retention (EPR) effect and CD44 endocytosis. The nanosheets showed excellent tumor growth inhibition with laser irradiation, indicating synergistic therapeutic efficacy, compared to a single treatment (i.e., with PTT alone or chemotherapy alone) [106].

HA-modified PB NPs loaded with 10-hydroxycamptothecin showed complete tumor growth inhibition, proving their synergistic effect, compared with either therapy alone Table 2. The targeting efficacy of HA was proven at in vivo temperatures, as observed in a HeLa tumor-bearing mice model, showing increases to 53 °C. In contrast, the temperature of tumors without HA modification only increased to 43 °C [107].

Few studies have explored MNP-based nanomaterials for combined hyperthermia and chemotherapy in the presence of AMF. One group [108] developed meso-2,3-dimercaptosuccinic acid-functionalized CoFe_2_O_4_ NPs with DOX (CoFe_2_O_4_@DMSA/DOX) for combined hyperthermic chemotherapy. The MHT efficiency of CoFe_2_O_4_@DMSA/DOX has been proven by AMF, showing the temperature of these NPs to reach 42 °C at a concentration of 125 μg/mL. Moreover, the combination of CoFe_2_O_4_@DMSA/DOX and MHT treatment showed significant cell cytotoxicity in MDA–MB–231 cancer cells, as compared to that of a single treatment. The CoFe_2_O_4_ NPs could be efficient and desirable nanomaterials for combined MHT. SPION, which have been recognized as MHT agents, have been recently utilized as photothermal agents as well [109,110]. One group [111] designed multi-functional polymeric micelles using HA, docetaxel (DTX), and SPION (HA–SPION–DTX) for combined photothermal chemotherapy. The HA–SPION–DTX could be specifically internalized into MDA–MB–231 cancer cells through HA–CD44 endocytosis. The photothermal effect of micelles was proven by NIR laser irradiation, showing a gradual increasing temperature of the micelles, according to increasing concentration. Moreover, HA–SPION–DTX with laser irradiation presented a synergistic therapeutic effect in MDA–MB–231 cancer cells. Thus, HA–SPION–DTX might be an effective therapeutic agent for combined hyperthermia and chemotherapy.

### 4.2. Combined Therapy with Photodynamic Therapy (PDT)

Photodynamic therapy (PDT) is a clinical cancer treatment in which a laser is used to activate light-absorbing molecules or photosensitizers to treat cancers [113,114]. After laser irradiation at the tumor site, cytotoxic singlet oxygen will be produced by activating the light-absorbing molecules or photosensitizers that are present in the tumor. The generated cytotoxic singlet oxygen from photosensitizer further triggers apoptotic and necrotic cancer cell death [115]. Thus, the effectiveness of PDT is determined by the generation of singlet oxygen [116]. Furthermore, PDT has been used to induce a significant immune response, which further induces the anti-tumor effect. In relation to hyperthermia, combined hyperthermia PDT approaches increase the therapeutic efficacy more than either treatment alone. Due to the effects of altering the cellular system by hyperthermia, introduction of PDT further affects tumor vasculature damage, cancer cell destruction, and inflammation, which can lead to a synergistic anti-cancer effect. Moreover, due to weak permeability or damage of cancer cell membranes induced by hyperthermia and PDT, the synergistic therapeutic effect can be accelerated [117,118]. Thus, combining hyperthermia with PDT can provide a great advantage in the therapeutic effect, compared to single treatment with either of the methods. Due to the ability for specific cancer targeting and the functionality of HA, hyperthermic agents and photosensitizers or light-absorbing molecules can be easily incorporated into HA.

In a recent study, a HA–GNR platform has been fabricated by Au–S bond synthesis. In this study, NPs were degraded by response in pH, glutathione (GSH), and HAase. After NIR laser irradiation, the released photosensitizer could be applied in PDT, so that the NPs elicited a combined cancer treatment [119]. Pharmacokinetic studies indicated long blood circulation times (*t*_1/2_ = 1.9 h) and, thus, enhanced tumor-targeted delivery, due to HA.

Single-walled carbon nanotubes (SWCNTs) coated with ICG-coupled HA NPs (IHNAPT) were synthesized for enhanced PTT and PDT. Due to the specific targeting ability of HA, these NPs were preferentially internalized into CD44-over-expressing SCC7 cancer cells. The photothermal effect of IHNAPT was proven in SCC7 tumor-bearing mice model, showing a tumor surface temperature of 55.4 ± 1.8 °C with NIR laser irradiation. Thus, IHNAPT could provide a synergistic anti-cancer effect by inhibiting tumor growth under laser irradiation, using both PTT and PDT [120].

An HA-based nanoplatform using PPY and IR-780 (IPPH) has been shown to be effective in combined PTT and PDT in SCC7 cancer cells. Under NIR irradiation, IPPH produces ROS and heat that synergistically leads to cancer cell death. In vivo anti-tumor studies have further demonstrated complete tumor ablation by PTT and PDT laser irradiation. Pheophorbide, a conjugated acetylated HA (AHP)-coated magnetite nanoparticle (AHP@MNPs), was designed for PTT/PDT combination therapy [77]. The AHP@MNPs demonstrated elevated generation of heat and singlet oxygen. Moreover, an enhanced anti-cancer effect was achieved in a k1735 tumor-bearing mice model, in response to a PTT and PDT synergistic system. Due to the excellent targeting effect of HA in cancer cells, PDT- and PTT-responsive agents simultaneously accumulate in cancer cells, following which external laser irradiation enables the combined treatment.

Carbon nanotubes (CNTs) have emerged as promising agents for PTT, due to their thermal conversion ability; they also have a high aspect ratio, large surface area, and surface chemical functionality. CNTs have, thus, shown great potential in various biomedical applications [121,122,123]. The hematoporphyrin monomethyl ether (HMME) PDT agent was adsorbed onto functionalized CNTs (HMME–HA–CNTs) for synergistic therapeutic efficacy in PTT and PDT [122]. Combined PTT with PDT using HMME–HA–CNTs showed superior anti-cancer effects, presenting significant tumor growth reduction when compared to that of PTT or PDT alone. Table 3 summarizes several studies on HA-modified nanoplatforms for photothermal PDT.

### 4.3. Combined Therapy with Immunotherapy

Immunotherapy is a type of cancer treatment, in which the immune system is used to recognize and attack cancer cells [125,126]. Dendritic cells (DCs) are the main antigen-presenting cells (APCs), which induce a T-cell-based immune response against cancer. The immunostimulatory effects of DC might be enhanced by hyperthermia through the up-regulation of IFN-γ secretion, which leads to the elimination of cancerous cells [127,128]. One study designed HA-based combination adjuvant systems by combining HA with immunostimulatory compounds (HA; Mw = 500–1300 kDa), showing anti-tumor therapeutic effects by prevention of tumor proliferation and a significant increase in the cytokine secretion of IL-6 and TNF-α [129].

Vaccines based on NP platforms have been fabricated by using HA immuno-adjuvants with NIR dyes (IR–7–lipo/HA–CpG, HA; Mw = 35 kDa) for combinatorial photothermal ablation and immunotherapy [130]. After NIR laser irradiation, the expression of CD40 and CD86, which are the hallmarks of DC maturation, showed significant upregulation. Additionally, elevated secretion of pro-inflammatory cytokines (IL-6 and TNF-α) was observed in IR–7–lipo/HA–CpG with laser irradiation. Therefore, IR–7–lipo/HA–CpG could have synergistic anti-cancer effects by use in combined PTT and immunotherapy.

### 4.4. Others

Radiotherapy (RT) is a type of cancer therapy which uses high-energy γ-rays or X-rays (photons) to generate sufficient free-radical damage or DNA damage (through the Fenton reaction) to damage solid tumors (or their vasculature) and induce cancer cell death. However, because of the poor radio-sensitivity and radio-resistance of tumors and the non-specificity of radiosensitizers, high doses of radiation are required to completely eliminate cancer cells [131,132]. Even though DNA damage is induced by RT, some cancer cells retain their inherent cancer tendency. In this cell stage, cancer cells are resistant to irradiation. In addition, DNA damage repair, cell cycle arrest, and tumor metabolism also cause radio-resistance. However, hyperthermia is known to hinder the repair of radiation-affected DNA damage [133,134]. Thus, combining hyperthermia with RT treatment elicits better anti-cancer effects. To enhance the concentrated X-ray dose in tumors, HA-conjugated NPs play a key role in increasing the local dose in the tumor region through CD44 endocytosis; thus, the X-ray dose in adjacent normal tissue/cells can be reduced [132].

HA-modified Au nanocages (AuNCs-HA, HA; Mw = 60,000~7000) were developed for RT + PDT + PTT therapy [135]. The in vitro synergistic effect was proven in a 4T1 cell cytotoxicity study. Either NPs with NIR or X-ray laser treatment alone presented cell viabilities at 74.2% and 40.1%, respectively; while the cell viability of NPs treated with NIR and X-ray decreased to 12.5%. Complete removal of the cancers was achieved for the 4T1 tumor-bearing mice model by NIR and X-ray irradiation, demonstrating potential candidates for RT + PDT + PTT.

## 5. Conclusions

This review shows the results of single and combined hyperthermic cancer treatments using HA-based nanomaterials, encapsulated with or conjugated to therapeutic agents for cancer treatments. To generate heat in the tumor regions, NIR laser or AMF are mainly utilized in hyperthermia-based cancer treatments. For a concentrated dose of therapeutic agents in cancer cells, HA-modified nanomaterials have been studied for their delivery potential. HA-modified nanomaterials could actively target cancer cells by CD44 receptor-mediated endocytosis and, thus, may elicit effective thermal efficacy through the use of photothermal materials, such as NIRs, AuNPs, GO, PB NPs, and SPIONs.

Due to the diverse alterations in response to hyperthermia, cancer cells are vulnerable to additional stimuli. Several trials of the combination of hyperthermic treatments with chemotherapy, PDT, and RT have further demonstrated the potential for the improvement of therapeutic efficacy. Thus, combining hyperthermic cancer treatment with HA has become a representative approach for synergistic cancer therapy, compared to the use of a single hyperthermic cancer treatment. HA-modified nanomaterials would be a smart starting line for the use of biomedical applications in hyperthermia cancer treatment.

Even though there are a number of HA nanoparticles and their abilities in targeting cancers have been reported, only a few of them are at the clinical stage. Complex synthesis processes also limit the industrialization of HA formulations. The lack of conjugation methods, as well as the excessive conjugation of different drugs or other moieties to HA, limits the receptor-mediated targeting of cancer cells. The molecular weight (Mw) of HA and its ability to target cancer cells are closely connected. A basic understanding of the metabolism of different Mw HA and the consequent targeting actions are yet to be explored; thus, more research needs to be done towards understanding the receptor-mediated uptake of HA nanoparticles according to size and Mw.

## Figures and Tables

**Figure 1 pharmaceutics-11-00306-f001:**
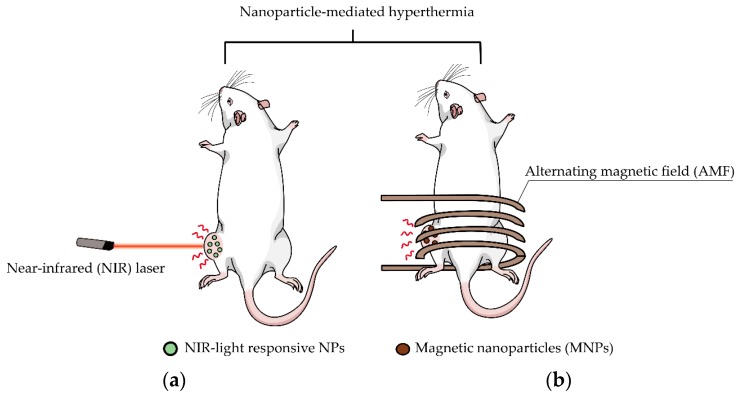
Schemes of nanoparticle (NP)-mediated hyperthermia. (**a**) Photothermal therapy (PTT) by near-infrared (NIR) laser irradiation to the tumor region, and (**b**) magnetic hyperthermia therapy (MHT) by alternating magnetic field (AMF) application.

**Figure 2 pharmaceutics-11-00306-f002:**
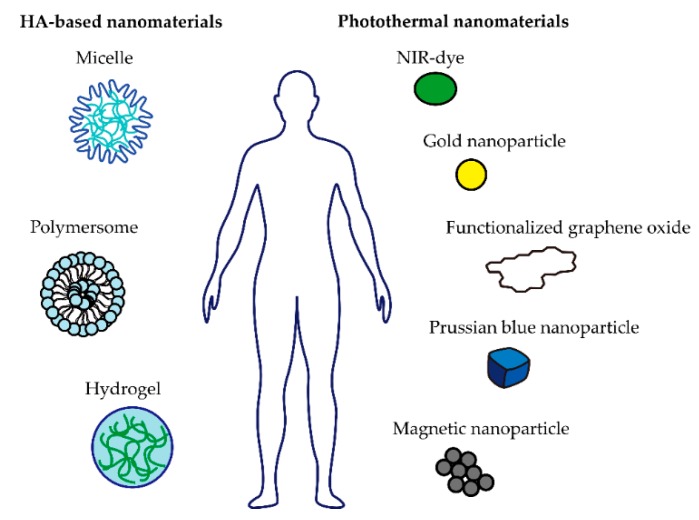
Schemes of hyaluronic acid (HA)-based and photothermal nanomaterials for hyperthermic cancer treatment.

**Figure 3 pharmaceutics-11-00306-f003:**
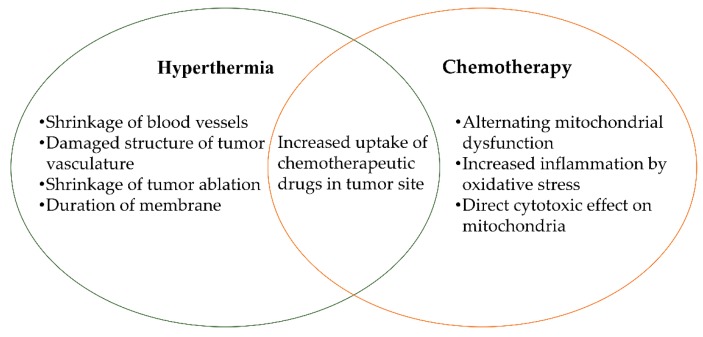
Aspects of combined photothermal chemotherapy cancer treatment.

**Table 1 pharmaceutics-11-00306-t001:** Nanomaterials for hyperthermic cancer treatment.

NIR-Responsive Materials	Molecular Weight of HA	Core	Composition	Cell Line	Status	Ref.
NIR-dye	0.48 MDa	IR780-iodide	HA-IR780	TC-1	In vivo	[12]
	50 kDa	IR808	HAIR	A549 (human lung carcinoma)	In vivo	[35]
	32 kDa	IR825	PFOB@IR825-HA-Cy5.5	HT-29	In vivo	[36]
Gold NPs	31.2 kDa	Fe_3_O_4_@Ag	Fe_3_O_4_@Au-HA NSs	HeLa (human cervical cancer)	In vivo	[42]
Graphene oxide NPs	100 kDa	Nano GO	NGO-HA	B16F10 (mouse melanoma cancer)	In vivo	[59]
	9.27 kDa	Reduced GO	HA-rGO	MCF-7, NHDF (normal human dermal fibroblast)	In vitro	[55]
Prussian blue NPs	5805 Da 31,200 Da	Fe_3_O_4_	Fe_3_O_4@_PB@PEI@BQDs-HA	HeLa	In vivo	[64]

**Table 2 pharmaceutics-11-00306-t002:** HA-modified nanoplatforms for photothermal chemotherapy.

NIR-Responsive Materials	Molecular Weight of HA	Chemotherapeutic Drugs	Cell Line	Status	Ref.
ICG	10 KDa	DOX	HCT-116	In vivo	[93]
Gold nanorods	8000 Da	DOX	MCF-7	In vivo	[96]
	190,000	DOX	SKOV-3 (human ovarian cancer), HOSEpiC (human ovarian surface epithelial cell)	In vitro	[99]
	5000 Da	DOX	MCF-7, MCF-7 ADR (drug-resistant human breast adenocarcinoma)	In vitro	[100]
Polypyrrole	200 kDa	Docetaxel (DTX)	4T1 (mouse breast cancer)	In vivo	[105]
PB NPs	32 k	10-hydroxycamptothecin	HeLa	In vivo	[107]
Graphene oxide	10,000	DOX	SKOV-3 (human ovarian cancer)	In vitro	[112]
	7000 kDa	DOX	A549	In vivo	[102]

**Table 3 pharmaceutics-11-00306-t003:** HA-modified nanoplatforms for combined photothermal photodynamic therapy (PDT).

NIR-Responsive Materials	Molecular Weight of HA	PDT Agents	Cell Line	Status	Ref.
Fe_3_O_4_	5.8 kDa	Pheophorbide-a (PheoA)	K1735 (murine melanoma), NIH3T3 (mouse embryonic fibroblast cell)	In vivo	[77]
IR-780 iodide	3 kDa	Polypyrrole	SCC7 (mouse head and neck squamous carcinoma), MDA-MB-231 (human breast cancer)	In vivo	[124]
Gold nanorods (GNRs)	8000 Da 170,000 Da	5-aminolevulinic acid (ALA)	MCF-7	In vivo	[119]
Carbon nanotubes (CNTs)	14,000-20,000	Hematoporphyrin monomethyl ether (HMME)	B16F10	In vivo	[122]

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
