# Peer review of "Biomedical Applications of Hyaluronic Acid-Based Nanomaterials in Hyperthermic Cancer Therapy"

_pharmaceutics, 2019, doi:10.3390/pharmaceutics11070306_

Reviewer 1 Report

The manuscript by Kim et al provides a review of hyaluronic acid-based nanomaterials in hypothermia therapy of cancer. The review includes nanoparticle-mediated hyperthermia and combination with other therapies. The topic is important and the manuscript is clearly organized. There’re a few comments for the authors to address.

Major comments:

It’s recommended a brief overview of manufacturing of HA-based nanoparticles (such as micelles, polymers and hydrogels) to be included.

Are there challenges or limitations in these HA-based approaches?

Minor comments:

Line 27, there’s no drugs or formulations having no side effects. Novel technologies can only have reduced side effects. The statement should be revised.

Molecular weights of HA should be provided in discussions or tables because it alters in vivo effects (e.g. ACS Biomater Sci Eng. 2015 Jul 13; 1(7): 481–493.).

More recent publications should be included and discussed including but not limited to:

Materials Science and Engineering: C, Volume 98, May 2019, Pages 419-436,

Acta Biomaterialia, Volume 83, 1 January 2019, Pages 400-413,

Mol. Pharmaceutics, Article ASAP, DOI: 10.1021/acs.molpharmaceut.9b00072.

Author Response

Respected professor,

Thank you so much for your valuable comments and suggestions. We have modified the manuscript according to your suggestions. Kindly go through it. All the comments were very helpful to improve the quality of our manuscript. Thanking you for your valuable time for correcting the review.

Kindly check the corrections I have made according to your suggestions.

Major comments:

It’s recommended a brief overview of manufacturing of HA-based nanoparticles (such as micelles, polymers and hydrogels) to be included.

: We modified the manuscript with the mention of brief overview and synthesis approach in HA-based NPs.

Line 53: “Due to biodegradability, biocompatibility, and nontoxicity of HA, diverse nanomaterial formations of HA such as micelles, polymersome, hydrogel, and nanoparticles could be designed to elicit cancer treatment.”

Line 102: “IR780-loaded HA-based micelles were synthesized via simple dialysis method for PTT effects in TC-1 lung cancer cells.”

Line 131: “This nanosystems were synthesized by a surfactant-free method….”

Line 377: “Recent studies fabricated HA-GNRs platform via Au-S bond synthesis.”

Are there challenges or limitations in these HA-based approaches?

: It is added the conclusion section.

In conclusions line 460: “Even though there are number HA nanoparticle and its ability in targeting cancer have been reported only few of them are in clinical stage. The complex synthesis process also limits the industrialization of HA formulations. In lack of conjugation method as well as the excessive conjugation of different drug or other moieties to HA limit the receptor mediated targeting to cancer cells. The molecular weight (Mw) of HA and its ability to target cancer cells are closely connected. Basic understanding of metabolism of different MW HA and its targeting actions yet to be explored. More studies have to be done to understand the receptor mediated uptake of HA nanoparticles according to size and MW”

Minor comments:

Line 27, there’s no drugs or formulations having no side effects. Novel technologies can only have reduced side effects. The statement should be revised.

: It is changed sentence and rewritten.

Line 27: “… to cancer site with reduced side effect.”

Molecular weights of HA should be provided in discussions or tables because it alters in vivo effects (e.g. ACS Biomater Sci Eng. 2015 Jul 13; 1(7): 481–493.).

: We have tried to add a molecular weight of HA in tables. All of tables were changed with colume of molecular weight. As well as, we changed the sentence with the mention of HA molecular weight

Line 109: “… (HA, Mw = 50 kDa) … “

Line 131: “This nanosystems were synthesized by a surfactant-free method and HA with molecular weight of 10 kDa were used”

Line 235: “… (Mw = 6800 Da) … “

Line 420: “…(IR-7-lipo/HA-CpG, HA; Mw = 35 kDa)…”

Line 440: “.... (AuNCs-HA, HA; Mw = 6000~7000) …”

More recent publications should be included and discussed including but not limited to:

Materials Science and Engineering: C, Volume 98, May 2019, Pages 419-436,

Acta Biomaterialia, Volume 83, 1 January 2019, Pages 400-413,

Mol. Pharmaceutics, Article ASAP, DOI: 10.1021/acs.molpharmaceut.9b00072.

: It is included and discussed recent publications.

We included and discussed recent publications.

Line 82: “….recent study reported a manganese phthalocyanine (MnPc) nanosheets as photoactive ligands and exhibited photothermal conversion efficiency of 72.3%..”

Line 162: “… good NIR absorbance, facile synthesis…” line 164, the number of 50 is its according reference.

Line 242: “HA tethered FePt alloy NPs were designed for multimodal therapy in glioblastoma cancer cell. In this study, due to the better magnetic property of FePt, magnetic hyperthermia led to 3-4 ℃ increase in cellular temperature, demonstrating superior heating ability.”

Line 308: “One group reported HA-modified and chemotherapeutic drugs loaded GO for targeted drug delivery and therapy. This group successfully loaded DOX and PTX onto GO-HA, showing significant cancer killing effect by MHT in CD44-overexpressing MDA-MB-231 cancer cells but not in BT-474 cells.”

Line 377: “Recent studies fabricated HA-GNRs platform via Au-S bond synthesis. In this study, NPs were degraded by response in pH, glutathione (GSH), and HAase. After NIR laser irradiation, released photosensitizer could be applied in PDT so that these NPs elicited combined cancer treatment.”

Thanks

Reviewer 2 Report

English needs to be checked throughout the manuscript.

The idea that HA has been often tagged to a drug or a drug carrier to improve drug delivery to CD44-overexpressing cancer cells to effectively suppress cancer growth due to its ability to specifically target CD44 is well known and has been also covered by the same authors in their review published in polymers journal few months ago. Can the authors explain what is new in this review? The idea of substituting chemotherapeutic drugs with other thermal or photosensitizing moieties used in Photothermal therapy (PTT) or magnetic hyperthermia therapy (MHT) does not add the fact that in both, targeting of the carriers or the particles is advantageous and hyaluronic acid can be used as a targeting moiety. This information was well already covered in the teams’ polymers review? What is new here and what would this review add to the literature? How as well this review will add to the literature compared to the below reference?

N. Vijayakameswara Rao, Hong Yeol Yoon, Hwa Seung Han, Hyewon Ko, Soyoung Son, Minchang Lee, Hansang Lee, Dong-Gyu Jo, Young Mo Kang & Jae Hyung Park (2016) Recent developments in hyaluronic acid-based nanomedicine for targeted cancer treatment, Expert Opinion on Drug Delivery, 13:2, 239-252, DOI: 10.1517/17425247.2016.1112374

The review is only telling a story of the studies done before but with no critical addition to what was already published in the team’s previous review. It would have been more enriching to have this review merged with the previously published one by the team to end up with one strong literature rather than a kind of repetition of the concept of targeting using HA but changing the carried or encapsulated moiety from chemotherapeutic drug to photo or thermo sensitizing agents.

The review failed to mention the combination of this approach with immunotherapy.

Author Response

Respected professor,

Thank you so much for your valuable comments and suggestions. Kindly go through it. All the comments were very helpful to improve the quality of our manuscript. Thanking you for your valuable time for correcting the review.

Kindly check the corrections I have made according to your suggestions.

; We also agree with your opinion, since both that review is about HA-based nanomaterials.

The previous review published in journal of polymers explains about utilization of HA nanomaterials for cancer therapy in general. We have given a small section where it explains about the HA nanomaterials targeted hyperthermia. HA CD44 targeting has been reported long back and researchers still utilize this particle for different kind for cancer therapy as a carrier as well as a targeting moiety.

The suggested review (Rao NV et al) is very general topic of utilization of HA nanoparticles for targeting and discussed HA-based drug delivery systems.

Here in this review,

We mainly focus on HA modified nanomaterials for hyperthermia cancer treatment. As the hyperthermia treatment modality gaining attention nowadays we have tried to include maximum information regarding the same.

Regarding to mention of combination therapy with immunotherapy, we newly add one paragraph of “Combined therapy with immunotherapy”.

Line 410: “4.3. Combined therapy with immunotherapy …. Immunotherapy is a one kind of cancer treatment, which use the body’s immune system to recognize and attack cancer cells. Dendritic cells (DCs) are the main antigen-presenting cells (APCs), which induce T-cell based immune response against cancer. Immunostimulatory effects of DC might be enhanced by hyperthermia through up-regulating IFN-γ secretion that lead to eliminate cancerous cells. One study has designed HA-based combination adjuvant systems by combining HA with immunostimulatory compounds (HA; Mw = 500 ~ 1300 kDa), showing anti-tumor therapeutic effect by prevention tumor proliferation and significant increase in cytokine secretion of IL-6 and TNF-α.

Vaccine based on NP platforms were fabricated by using HA-immunoadjuvant with NIR dye (IR-7-lipo/HA-CpG, HA; Mw = 35 kDa) for combinatorial photothermal ablation and immunotherapy. After NIR laser irradiation, the expression of CD40 and CD86, which are the hallmark of DCs maturation showed significantly upregulated. As well as, elevated secretion of pro-inflammatory cytokines (IL-6 and TNF-α) was observed in IR-7-lipo/HA-CpG with laser irradiation group. Moreover, IR-7-lipo/HA-CpG could have synergistic anti-cancer effect by PTT and immunotherapy.

Thanks

Reviewer 3 Report

Authors presented and compiled good work on hyperthermia related work on Nano medicine. Since the entire work is focused on  hyaluronic acid-based nanomaterials the outcome was somewhat limited. I would ask authors to pay more attention how the HA based nanoparticles improve to target certain organs how that can be explored as targeted hyperthermia? Also, why and how HA based nanoparticles are unique in comparison to other nanos? All PK-PD and bioavailability data may be used to insert in one section.

Author Response

Respected Professor,

Thank you so much for your valuable comments and suggestions. We have modified the manuscript according to your suggestions. Kindly go through it. All the comments were very helpful to improve the quality of our manuscript. Thanking you for your valuable time for correcting the review.

Kindly check the corrections I have made according to your suggestions.

; Regarding to comments of HA targeting to hyperthermia, the CD44 targeting is not related to the hyperthermia effect of the nanoparticle. CD44 targeting only increase the accumulation of the nanoparticle in the site of action. Several kinds of PTT/AMF agents can be used for hyperthermia. Here the role of HA is just to target (targeting carrier) to the desired site for the delivery of drugs or hyperthermia agents.

We have included the targeting ability as well as other unique properties of HA in the introduction section with new references.

Line 37: “Hence naturally occurring polysaccharides are biodegradable HA is reported to be highly biodegradable. In addition to this as it is a polysaccharide its biocompatibility also makes it unique from other polymeric systems. HA did not show any immune response to the body and can be used as an efficient delivery system for cancer treatment.”

Line 46: “This CD44 targeting is unique property of HA-based nanomaterials.”

Line 47: “HA is widely distributed in body such as extracellular matrix and synovial fluids. HA mainly degraded by enzymatic process. Three different types of enzymes participate in HA degradation: hyaluronidase, b-d-glucuronidase and β-N-acetyl-hexosaminidase. Firstly, HA is degraded by hyaluronidase that split high molecular weight HA into small oligosaccharides. Subsequently, involvement of b-d-glucuronidase and β-N-acetyl-hexosaminidase help to degrade a remained oligosaccharide. The high biocompatibility, CD44 targeting ability as well as enzyme degradation makes HA-based nanoparticles unique from other nanoparticles.

We modified the sentence with data of PK-PD.

Line 133: “In compared to without HA coating NPs, GNS/siHSP72/HA showed improved pharmacokinetics with high accumulation in tumor.”

Line 380: “Pharmacokinetic studies indicated long blood circulation time (t1/2 = 1.9 h), meaning that enhanced tumor targeted delivery due to HA.”

Thanks.

Round  2

Reviewer 2 Report

No more comments. The manuscript should be checked better for typos and grammatical errors.